# The Mechanisms of Thiosulfate Toxicity against *Saccharomyces cerevisiae*

**DOI:** 10.3390/antiox10050646

**Published:** 2021-04-22

**Authors:** Zhigang Chen, Yongzhen Xia, Huaiwei Liu, Honglei Liu, Luying Xun

**Affiliations:** 1State Key Laboratory of Microbial Technology, Shandong University, 72 Binhai Road, Qingdao 266237, China; chenbio@sdu.edu.cn (Z.C.); xiayongzhen2002@sdu.edu.cn (Y.X.); liuhuaiwei@sdu.edu.cn (H.L.); 2School of Molecular Biosciences, Washington State University, Pullman, WA 991647520, USA

**Keywords:** *Saccharomyces cerevisiae*, mitochondria, thiosulfate, sulfite, elemental sulfur, sulfide

## Abstract

Elemental sulfur and sulfite have been used to inhibit the growth of yeasts, but thiosulfate has not been reported to be toxic to yeasts. We observed that thiosulfate was more inhibitory than sulfite to *Saccharomyces cerevisiae* growing in a common yeast medium. At pH < 4, thiosulfate was a source of elemental sulfur and sulfurous acid, and both were highly toxic to the yeast. At pH 6, thiosulfate directly inhibited the electron transport chain in yeast mitochondria, leading to reductions in oxygen consumption, mitochondrial membrane potential and cellular ATP. Although thiosulfate was converted to sulfite and H_2_S by the mitochondrial rhodanese Rdl1, its toxicity was not due to H_2_S as the *rdl1*-deletion mutant that produced significantly less H_2_S was more sensitive to thiosulfate than the wild type. Evidence suggests that thiosulfate inhibits cytochrome c oxidase of the electron transport chain in yeast mitochondria. Thus, thiosulfate is a potential agent against yeasts.

## 1. Introduction

Thiosulfate is spontaneously produced by reacting sulfite with elemental sulfur [1], and it is a key intermediate in the biogeochemical cycle of sulfur [2]. Many heterotrophic bacteria oxidize H_2_S to thiosulfate as a detoxification mechanism [3,4,5], and other bacteria oxidize thiosulfate to sulfate to gain energy for growth [6,7,8]. Further, bacteria and yeast readily use thiosulfate as a sulfur source for growth [9,10,11,12]. We recently characterized the pathway of thiosulfate assimilation in *Saccharomyces cerevisiae* [10]. Yeast uses sulfate transporters to uptake thiosulfate and then uses a mitochondrial rhodanese Rdl1 to transfer a zero-valence sulfur from thiosulfate to glutathione (GSH), producing glutathione persulfide (GSSH). GSSH spontaneously reacts with another GSH to release H_2_S and glutathione disulfide (GSSG) [13]. The produced H_2_S is used for the production of cysteine and methionine [10,14]. 

Under acidic conditions, thiosulfate spontaneously breaks into sulfurous acid (H_2_SO_3_) and elemental sulfur (S^0^) [15]. Both sulfite and S^0^ are toxic to yeast. The inhibitory effect of sulfite on yeast has been extensively investigated, as sulfite is widely used in fruit and vegetable preservation [16,17] and in ethanol fermentation to inhibit the growth of wild yeasts [18,19]. Undissociated H_2_SO_3_ at low pH is the most effective species as it kills yeasts [20]. At neutral pH, sulfite is also inhibitory because it decreases ATP production by inhibiting glyceraldehyde-3-phosphate dehydrogenase (GAPDH) and alcohol dehydrogenase in *S. cerevisiae* [21,22]. Sulfite does not inhibit the electron transport chain in yeast [23]. Further, sulfite can be oxidized by reactive oxygen species to generate sulfur trioxide free radicals (HSO_3_^−^ and SO_3_^−^) [24,25], which may damage DNA and destroy tryptophan [26,27]. S^0^ is a common fungicide [28,29]. S^0^ is believed to be transported into the cell as hydrogen polysulfide (H_2_S_n_) [30,31], which reacts with cellular thiols, including protein thiols and GSH, to form organic persulfide and polysulfide (RS_n_H, _n_ ≥ 2). GSH spontaneously reacts with RS_n_H to produce H_2_S and GSSG, and S^0^ toxicity could be due to the combination of thiol modification and H_2_S production [32]. In addition, RS_n_H may inhibit enzymes with metal ions or heme in their active centers [33]. 

Thiosulfate is a relatively benign sulfur species, and it is commonly used as a remedy to treat cyanide poisoning [34]. Its toxicity to microorganisms has not been reported to date. Here, we report that thiosulfate is more inhibitory than sulfite to the yeast. At low pH, thiosulfate is converted to S^0^ and H_2_SO_3_, both of which are toxic; at neutral pH, thiosulfate is inhibitory by itself.

## 2. Materials and Methods

### 2.1. Materials and Reagents

Sodium thiosulfate, sodium sulfite, sulfur powder (S^0^), glyceraldehyde 3-phosphate, N, N, N′, N′-Tetramethyl-p-phenylenediamine dihydrochloride (TMPD), ascorbate and nicotinamide adenine dinucleotide (NAD^+^) were purchased from Sigma Chemical (St. Louis, MO, USA). The enzymes used for DNA manipulations were obtained from Thermo Fisher (Waltham, MA, USA). PCR enzymes were purchased from Toyobo (Osaka, Japan). The RNA extract kit (R6834-01) was purchased from Omega (Norcross, GA, USA). Excess sulfur powder was added to acetone to obtain saturated sulfur in acetone (S^0^, 20 mM).

### 2.2. Strains, Mutants, and Plasmids 

General cloning and site-directed mutagenesis were performed by using previously reported methods [35,36]. Sequencing and PCR were carried out according to standard procedures. The yeast strain used in this study was *S. cerevisiae* BY4742 (*MATα his3*Δ*1 leu2*Δ*0 lys2**Δ**0 ura3*Δ*0*). The *Escherichia coli* strain DH5α served as the host strain for all plasmid constructions, and it was grown in lysogeny broth (LB) medium (0.5% yeast extract, 1% peptone, and 1% NaCl). When required, 50 μg/mL ampicillin was added to LB medium. *S. cerevisiae* strains were grown at 30 °C in yeast extract-peptone-dextrose (YPD) medium (1% yeast extract, 2% peptone, and 2% glucose) or synthetic defined (SD) medium (0.17% YNB, 5% (NH_4_)_2_SO_4_, 2% glucose, amino acid mixtures) supplemented with auxotrophic requirements. The pH of the SD medium (liquid and plate) was routinely adjusted to 6 by adding NaOH. One-step PCR-mediated gene disruption was carried out to delete *RDL1* (NP_014928.1) and *RDL2* (NP_014929.3) in BY4742 [37]. The recombinant plasmids were constructed using a previously reported method [10]. The strains and plasmids used in this study are listed in Table 1. All primers are listed in Table 2. 

### 2.3. Measurement of Growth Curves of S. cerevisiae in the Presence of Thiosulfate or Sulfite

Fresh cells of *S. cerevisiae* were inoculated in 5 mL of SD medium and grown overnight at 30 °C with shaking at 200 rpm. Cells were then centrifuged at 10,000× *g* for 5 min, and the pellet was resuspended in SD medium. Equal amounts of cells (OD_600nm_ = 0.1) were cultured in 400 μL of SD medium containing additional sulfite, thiosulfate or S^0^ at 30 °C with shaking; the growth was measured at OD_600nm_ by using a microplate reader (BioTek, Synergy H1). The V_max_ of yeast cells growth in SD medium with different amount of thiosulfate was calculated during the middle log phase, which was used for analyzing the IC_50_ values of thiosulfate.

Thiosulfate and sulfite tolerance assays were performed on SD agar plates (SD medium with 1.5% agar). The cells of overnight culture in SD medium were serially diluted with fresh SD medium, and 5 µL of each dilution was spotted on the SD agar plates containing thiosulfate or sulfite. The plates were incubated at 30 °C for 48 h before scanning with a scanner (Fluor Chem Q Alpha, Santa Clara, CA, USA).

The pH of SD medium (liquid and plate) was adjusted from 6 to 4 and 5 by adding HCl when needed.

### 2.4. Detection of Killing Effect of Thiosulfate at Low pH 

Fresh cells of the *S. cerevisiae* strain were inoculated in 5 mL of SD medium and grown overnight at 30 °C with shaking at 200 rpm. Cells were collected by centrifugation (10,000× *g*, 5 min) and suspended in water. Equal amounts of cells (OD_600nm_ = 1) were suspended in citric acid–sodium phosphate dibasic buffer (CPBS, pH 3.4, 4, 5, 6) with or without 10 mM thiosulfate and incubated at 30 °C for 1 h. The suspensions were then diluted 1:10,000 times with sterile water. Dilutions (100 μL) were added to the YPD plate and incubated at 30 °C for 2 days. Colony-forming units (CFUs) were counted.

The yeast cells incubated in CPBS buffer (pH 3.4) were centrifuged (10,000× *g*, 5 min), and the pellet was suspended in HEPES buffer (50 mM, pH 7.0). The yeast cells were observed under an Olympus microscope (IX83, Olympus, Tokyo, Japan).

### 2.5. Rhodanese Assay 

The standard rhodanese assay was conducted in the same manner as previously reported [3]. Briefly, fresh yeast cells were collected, washed twice with water, an amount corresponding to an OD_600nm_ of 2 was suspended in ice-cold 50 mM Tris-HCl (pH 8) and disrupted using a pressure cell homogenizer (Stansted Fluid Power LTD SPCH-18, Harlow, UK). The suspension was centrifuged at 12,500× *g* for 10 min to remove cell debris, and the protein concentration in cell lysate was measured by using a microspectrophotometer (Bio future, K5500). An assay volume of 1 mL included 50 mM Tris-HCl (pH 8), 5 mM sodium thiosulfate, 10 mM KCN and cell lysate. Reactions were initiated by the addition of KCN and terminated after 10 min by boiling for 3 min. Subsequently, 0.1 mL of ferric nitrate reagent was added and centrifuged at 12,500× *g* for 5 min; the absorbance was measured at 460 nm and compared with a standard curve of thiocyanate. 

### 2.6. Determination of Cellular Thiosulfate Concentration

The sulfur-starved yeast cells were prepared and resuspended at an OD_600nm_ of 10 in sterile phosphate buffered solution (PBS) buffer containing 2% glucose, as previously reported [10]. A final concentration of 200 μM of thiosulfate was added, and the cells were incubated at 30 °C for 60 min. Cells were centrifuged (13,000× *g*, 5 min), washed twice with water, resuspended in ice-cold 50 mM Tris-HCl (pH 8) to an OD_600nm_ of 5 and disrupted using a pressure cell homogenizer. The cell lysate was again centrifuged at 12,500× *g* for 10 min to remove cell debris and thiosulfate in the supernatant was derivatized with monobromobimane (mBBr) and detected as previously described [10]. Thiosulfate, sulfite and sulfide react with mBBr to produce derivatives that can be separated by HPLC and detected with a fluorescence detector [38]. The cellular concentration of thiosulfate was calculated using a reported haploid cell volume of 50 fL [39,40].

### 2.7. Measurements of Mitochondrial Membrane Potential and Cellular ATP Concentration

Overnight yeast cells were collected and suspended in SD medium at an OD_600nm_ of 1 with or without 10 mM thiosulfate, and the samples were incubated at 30 °C for 1 h. Yeast cells were collected, washed twice with ice-cold HEPES buffer (50 mM, pH 7.0) and suspended in the same buffer to an OD_600nm_ of 2. The cells were used to measure mitochondrial membrane potential by using a fluorescent probe 5,5’,6,6’-tetrachloro-l,l’,3,3’-tetraethylbenzimidazolo-carbocyanine iodide (JC-1) (Beyotime Biotech, Shanghai, China) [41], following the manufacturers’ instructions. Briefly, 1 mL of the yeast cell suspension was mixed with an equal volume of the purchased JC-1 staining solution (5 μg/mL), incubated at 30 °C for 20 min, rinsed twice with ice-cold HEPES buffer (50 mM, pH 7.0) and suspended in the ice-cold HEPES buffer at OD_600nm_ of 0.5. The fluorescence (red fluorescence: Ex = 556 nm and Em = 590 nm; green fluorescence: Ex = 490nm and Em = 530 nm) was measured by using a microplate reader (BioTek, Synergy H1). The thiosulfate-treated and untreated yeast cells were tested. In theory, JC-1 exists as monomers that emit green fluorescence in cells with low mitochondrial membrane potential, and it forms aggregates that emit red fluorescence in cells with high mitochondrial membrane potential. A decrease in the ratio of red to green fluorescence indicates a reduction in the mitochondrial membrane potential [42]. 

For ATP determination, the cells were disrupted, and the lysate was centrifuged at 12,500× *g* for 10 min to remove cell debris. The protein concentration in the supernatant was measured by using a microspectrophotometer (Bio future, K5500). The ATP concentration was detected using firefly luciferase, and the produced chemiluminescence was measured by using a plate reader (BioTek, Synergy H1) [43]. 

### 2.8. Assaying GAPDH Activity

The assay of GAPDH activity was carried out according to a reporter method with minor modification [44]. Fresh yeast cells were collected by centrifugation, washed twice with distilled water, suspended in ice-cold lysis buffer (40 mM triethanolamine, 50 mM Na_2_HPO_4_, 5 mM EDTA; pH 8.6) to an OD_600nm_ of 2, and disrupted using a pressure cell homogenizer. The lysate was centrifuged at 12,500× *g* for 10 min to remove cell debris, and the protein concentration was measured by using a microspectrophotometer (Bio future, K5500). The lysate was incubated with 10 mM thiosulfate or 10 mM sulfite at 30 °C for 30 min. A control without thiosulfate and sulfite was also incubated. Then, the treated lysate containing 0.5 mg of protein was incubated with 1.5 mM glyceraldehyde 3-phosphate and 1 mM NAD^+^ in 1 mL of the lysis buffer. The generation of NADH was analyzed by recording the absorbance increase at 340 nm for 30 s. The rate of NADH production was calculated.

### 2.9. Measuring Oxygen Consumption

The sulfur-starved yeast cells were prepared and resuspended in sterile PBS buffer containing 2% glucose at an OD_600nm_ of 10, as previously reported [10]. Thiosulfate was added to a final concentration of 10 mM, and oxygen was monitored by using an Orion RDO meter (Thermo Scientific Inc., Waltham, WA, USA). The RDO meter was calibrated with air-saturated water according to the manufacturers’ instructions. Yeast cells without thiosulfate addition were used as the control. 

The sulfur-starved yeast cells were resuspended in sterile PBS buffer containing 12.5 mM ascorbate and 1.4 mM N, N, N′, N′-tetramethyl-p-phenylenediamine (TMPD) at an OD_600nm_ of 10. Thiosulfate or sulfite was added to a final concentration of 10 mM, and oxygen was monitored by using the Orion RDO meter (Thermo Scientific Inc., Waltham, WA, USA). Yeast cells without thiosulfate addition were used as the control. 

## 3. Results

### 3.1. Thiosulfate Inhibited the Growth of S. cerevisiae at pH 6 and Killed It at pH 3.4

The growth of *S. cerevisiae* in SD medium was inhibited by high concentrations of thiosulfate (Figure 1A) and sulfite (Figure 1B). Surprisingly, thiosulfate was more inhibitory than sulfite (Figure 1C). The pH of fresh SD medium was approximately 6.0, and the pH decreased to about 2.1 after cultivating the yeast. 

To test whether pH affected the tolerance of *S. cerevisiae* to thiosulfate, a series of SD mediums with different starting pHs were used. Yeast cells grew well in SD mediums adjusted to various pH values. When 10 mM thiosulfate was added to the medium, *S. cerevisiae* displayed reduced growth as the pH dropped, suggesting that thiosulfate has pH-dependent effects on *S. cerevisiae* (Figure 2A). The pH-dependent inhibitions also occurred on agar plates (Figure 2B).

Colony forming units (CFUs) were measured after incubation with 10 mM thiosulfate for 60 min in citric acid–sodium phosphate dibasic buffer at different pH values. CFUs did not decrease at pH 4.4 to 6.0, but CFUs sharply decreased at pH < 4 (Figure 2C). At pH 3.4, thiosulfate killed almost all *S. cerevisiae* cells with erupted cells observed under microscopy (Figure 2D). The results indicate that thiosulfate mainly inhibits the yeast growth at slightly acidic pHs but kills the yeast under acidic conditions. 

HPLC analysis showed that approximately 8% of 10 mM thiosulfate decomposed at pH 3.4 in 1 h, likely producing 0.8 mM sulfite and 0.8 mM S^0^ as previously reported [45]. When yeast cells (OD_600nm_ = 1) in 1 mL of citric acid–sodium phosphate dibasic buffer were incubated with sulfite, S^0^ or thiosulfate at pH 3.4 and 30 °C for 1 h, sulfite killed almost all the yeast cells and 0.8 mM sulfur powder killed approximately 2/3 of the yeast cells, but 0.8 mM thiosulfate did not have any apparent lethal effect on the yeast cells (Appendix A). At pH 6, 0.8 sulfite and thiosulfate did not reduce CFUs, but 0.8 mM S^0^ slightly reduced CFUs (Appendix A). These results indicate that at acidic pHs thiosulfate is slowly decomposed to S^0^ and H_2_SO_3_ that kill yeast cells. 

### 3.2. Thiosulfate Is Actively Transported into the Cells for Its Inhibition

Sul1, Sul2 and Soa1 are the three main permeases responsible for sulfate and thiosulfate uptake in *S. cerevisiae*, and a mutant with the three genes deleted (triple mutant) was found to be deficient in thiosulfate uptake [10]. The triple mutant yeast grew better than the wild type in SD medium with 10 mM thiosulfate (Figure 3), suggesting that thiosulfate is actively transported into yeast cells where it exhibits its toxicity.

### 3.3. Thiosulfate Itself Is Inhibitory to the Yeast at pH 6 

Inside the yeast, the mitochondrial rhodanese, Rdl1, is mainly responsible for converting thiosulfate to H_2_S [10], and its deletion reduced the whole-cell rhodanese activity by half (Table 3). The *RDL1* mutant was more sensitive to thiosulfate than the wild type (Figure 4A). Complementation of Rdl1 to the **Δ*rdl1* mutant restored the relative resistance, but complementing the mutant protein, Rdl1-C98S, without rhodanese activity to the **Δ*rdl1* mutant did not alleviate thiosulfate inhibition (Figure 4A). The IC_50_ values of thiosulfate were 9.45 mM and 4.74 mM for the wild type and **Δ*rdl1* mutant, respectively (Figure 4B). When sulfur-starved yeast cells were incubated with 0.2 mM thiosulfate in 50 mM potassium phosphate (pH 6) containing glucose for 1 h, the wild type and **Δ*rdl1* mutant accumulated ~0.9 mM and ~1.6 mM thiosulfate inside the cell, respectively. During the incubation, the wild type released large amounts of H_2_S, whereas the **Δ*rdl1* mutant did not, and the released H_2_S was detected using a lead acetate paper strip affixed in the gas phase of the tube (Appendix A). When another rhodanese, Rdl2, was deleted, the **Δ*rdl2* mutant and the wild type showed similar resistance to thiosulfate (Appendix A); however, the **Δ*rdl1 *Δ*rdl2* double mutant showed slightly more sensitive to thiosulfate than the **Δ*rdl1* mutant. Overexpression of *RDL2* in the **Δ*rdl1* strain did not restore the relative tolerance of the **Δ*rdl1* strain to thiosulfate (Appendix A), perhaps due to variations in subcellular location. These results collectively show that thiosulfate is inhibitory to the yeast growth at pH 6 and Rdl1 alleviates thiosulfate toxicity by actively converting thiosulfate into H_2_S and sulfite. Further, the produced H_2_S is not inhibitory to the yeast. 

### 3.4. Thiosulfate Perturbs the Mitochondrial Bioenergetics in S. cerevisiae

A sharp drop in cellular ATP concentration occurred after the addition of thiosulfate to yeast cells (Figure 5A). Thiosulfate also caused a decrease in the mitochondrial membrane potential (Figure 5B). These results indicate that thiosulfate perturbs the cellular bioenergetics. Since sulfite inhibits the activity of GAPDH [23], the effect of thiosulfate on the enzyme was tested. Thiosulfate did not inhibit the enzyme activity, but the sulfite control did (Figure 5C).

Copper (Cu^2+^) is an essential trace metal of the electron transport chain of yeast mitochondria [46]. When thiosulfate reacted with Cu^2+^, an intense color change occurred (Appendix A), likely due to the reduction of Cu^2+^ to Cu^+^ by thiosulfate as previously reported [47]. Thiosulfate also severely inhibited oxygen consumption by yeast cells (Figure 5D). TMPD and ascorbate are often used to donate electrons to cytochrome c, which is then oxidized by cytochrome c oxidase at the expense of O_2_ [48]. Replacing glucose with ascorbate/TMPD, thiosulfate still inhibited oxygen consumption, but sulfite did not (Figure 5E). These results suggested that excessive thiosulfate interferes with the function of the electron transport chain in mitochondria, likely by targeting Cu^2+^ in cytochrome c oxidase. 

## 4. Discussion

Thiosulfate is toxic to *S. cerevisiae*. The toxic mechanisms are different at neutral or acidic pHs (Figure 6). At an acidic pH, it slowly decomposes to S^0^ and sulfurous acid (Rolia et al., 1982), and in our test about 8% of 10 mM thiosulfate decomposed at pH 3.4 in 1 h. Both S^0^ and sulfurous acid are highly toxic to the yeast [20,29]. The decomposition likely occurs in the medium, as the pH inside yeast cells is relatively neutral even under acidic conditions [49]. Near neutral pH, thiosulfate is actively transported into the cells and accumulated inside the cell (Figure 3) [10]. The accumulated thiosulfate directly inhibits O_2_ consumption, which in turn lowers membrane potential and ATP levels (Figure 5). Yeast rhodaneses, primarily Rdl1, convert thiosulfate to H_2_S and sulfite, alleviating thiosulfate inhibition (Figure 4). The proposed mechanism of thiosulfate toxicity is summarized in Figure 6. 

The direct target of thiosulfate is likely the electron transport chain (Figure 5), which is also the target of H_2_S. The cytochrome c oxidase of the mitochondrial electron transport chain contains two copper centers Cu_A_ and Cu_B_ that are involved in electron transfer [50]. Sulfide binds to Cu_B_, making its re-oxidation difficult and blocking the electron flow [51]. Although thiosulfate is slowly converted into sulfite and H_2_S by Rdl1 in yeast [10], the produced H_2_S is likely not high enough to inhibit the yeast as the *Δrdl1* mutant that produces much less H_2_S is more sensitive to thiosulfate (Figure 4A). Since both thiosulfate and sulfide are able to reduce Cu^2+^ to Cu^+^ (Appendix A) [47], thiosulfate also has the potential to bind to Cu_B_ to block the electron flow. The activity of cytochrome c oxidase can be directly assayed in whole cells with artificial electron donor TMPD, which is again reduced by ascorbate [48]. In the assay, thiosulfate inhibited cytochrome c oxidase (Figure 5E). This finding explains why thiosulfate decreased the oxygen consumption, mitochondrial membrane potential and ATP levels in *S. cerevisiae* (Figure 5). 

The effect of thiosulfate is different from that of sulfite. Sulfite is widely used in fruit and vegetable preservation [16,17]. It is highly toxic at low pH, as undissociated H_2_SO_3_ is the effective agent, which easily kills yeasts [20]. At relatively neutral pH, sulfite is toxic only at high concentrations and is a known inhibitor of GAPDH, slowing down glycolysis and ATP production in *S. cerevisiae* [22,52,53]. However, sulfite does not inhibit oxidative phosphorylation in the mitochondria [23].

S^0^ is commonly used as a fungicide and employed widely in traditional agriculture as an eco-friendly fungicide to protect vineyards against *Botrytis cinerea* [54]. S^0^ exhibited antimicrobial activity (MIC = 5.47 µg/mL) against *Staphylococcus aureus* and had inhibitory effects on membrane lipids of *Aspergillus niger* [55,56]. However, its efficiency is impaired by its low solubility [28,29]. To circumvent the limitation, soluble organosulfur compounds that release S^0^ have been synthesized and used to treat antibiotic-resistant bacteria [57]. Thiosulfate may be used as a soluble inorganic source of S^0^ under acidic conditions to inhibit yeast and pathogenic microorganisms. A US patent has used a *Lactobacillus* strain with thiosulfate to treat urogenital yeast infections [58]. Although the mechanism is not reported, we can speculate that the *Lactobacillus* strain produces lactic acid and lowers pH that facilitates thiosulfate decomposition to release S^0^ and H_2_SO_3_. Thus, at an acidic pH thiosulfate is a good source of S^0^ and H_2_SO_3_, both of which are effective agents against *S. cerevisiae*. 

## 5. Conclusions

Thiosulfate is an inhibitor of *S. cerevisiae*. It is more toxic than sulfite at near neutral pH, and it directly inhibits the electron transport chain. The inhibition is reversible, as the yeast recovered in fresh media without thiosulfate. The finding does not contradict the use of thiosulfate in treating cyanide poisoning in humans [34] since high doses of thiosulfate may only temporarily inhibit aerobic respiration. Thiosulfate should be rapidly released in the urine, and it has been shown to be safe in animal studies [59]. At an acidic pH, thiosulfate is a good source of S^0^ and H_2_SO_3_, both of which kill yeast (Figure 6). The finding may guide the appropriate use of thiosulfate as a preservative for foods or fruits or as an agent against pathogenic microorganisms. 

## Figures and Tables

**Figure 1 antioxidants-10-00646-f001:**
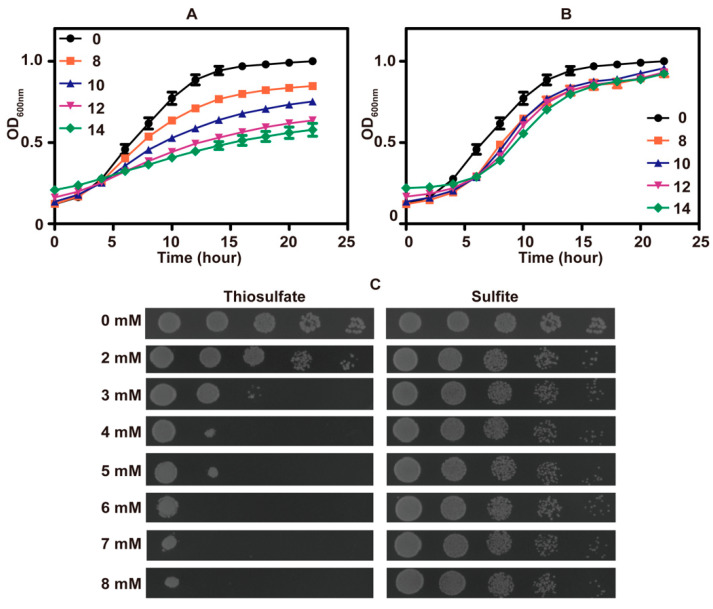
The phenotype analysis of *S. cerevisiae* in SD medium. The growth curves of *S. cerevisiae* in SD medium (pH 6) with different amounts of thiosulfate (**A**) or sulfite (**B**); the concentrations of thiosulfate or sulfite are in mM. The data are averages with standard deviations from three replicates. (**C**) The thiosulfate or sulfite tolerance assay on SD agar (pH 6) plates.

**Figure 2 antioxidants-10-00646-f002:**
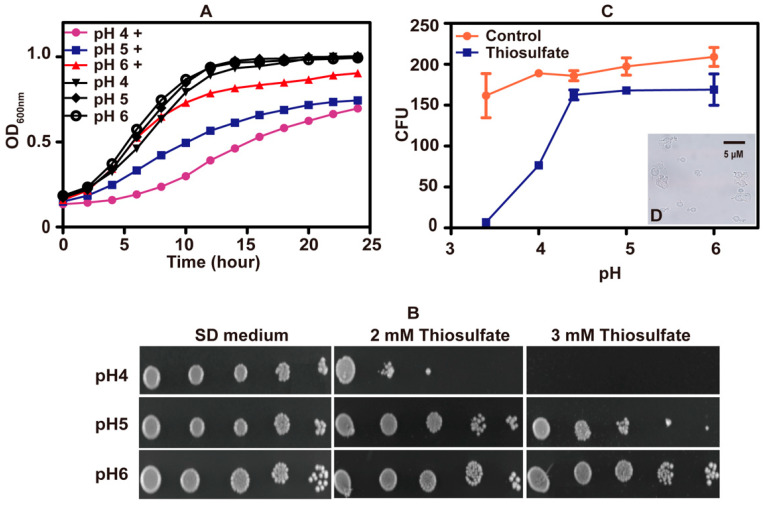
The toxicity effect of thiosulfate changed with pH. (**A**) The growth curves of *S. cerevisiae* in SD medium at different pH. “+” label indicates “with 10 mM thiosulfate”. (**B**) The tolerance of *S. cerevisiae* to thiosulfate on SD agar plates of different pH. (**C**) CFUs of *S. cerevisiae* after incubating for 1 h in citric acid–sodium phosphate dibasic buffer of different pH (3.4, 4, 4.4, 5 and 6) with 10 mM thiosulfate or without thiosulfate (control). The growth curves are averages of three samples in the 48-well plate (Figure 2A); the data in Figure 2C are averages of three samples with standard deviations. (**D**) (Figure 2C insert) The image of lysed *S. cerevisiae* cells after incubating in citric acid–sodium phosphate dibasic buffer (pH 3.4) with 10 mM thiosulfate for 1 h.

**Figure 3 antioxidants-10-00646-f003:**
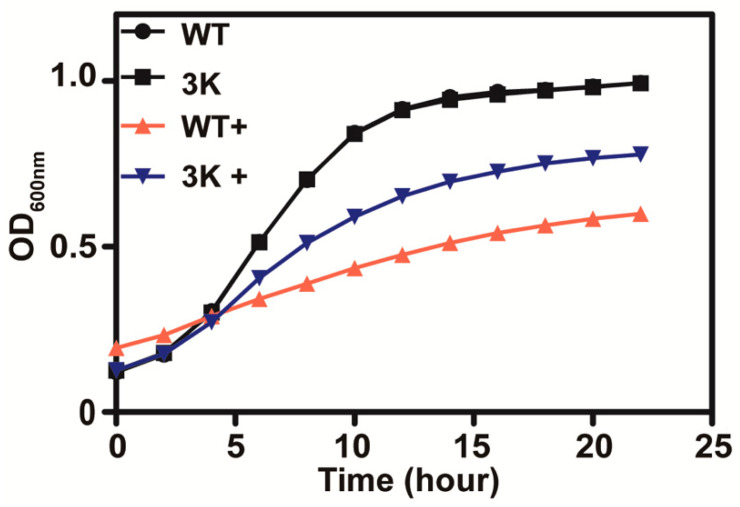
The thiosulfate transporter mutant (**Δ*sul1 *Δ*sul2 *Δ*soa1*) is less sensitive to thiosulfate in SD medium. The presence of 10 mM thiosulfate is indicated with “+”. The curves of the wild type and the 3K mutant without thiosulfate overlap. The growth curves are averages of three samples in 48-well plates.

**Figure 4 antioxidants-10-00646-f004:**
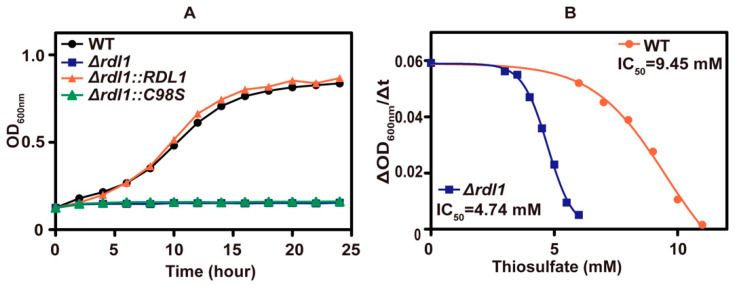
The tolerance of different *S. cerevisiae* strains to thiosulfate. (**A**) The growth curves of different *S. cerevisiae* strains in SD medium containing 8 mM thiosulfate. (**B**) The IC_50_ of thiosulfate in yeast cells. The V_max_ of yeast cells growth was calculated in the middle log phase and used for analyzing IC_50_ of thiosulfate. The growth curves are averages of three replicates in 48-well plates. The P values (by t test) for the **Δ*rdl1* versus wild type were all <0.01.

**Figure 5 antioxidants-10-00646-f005:**
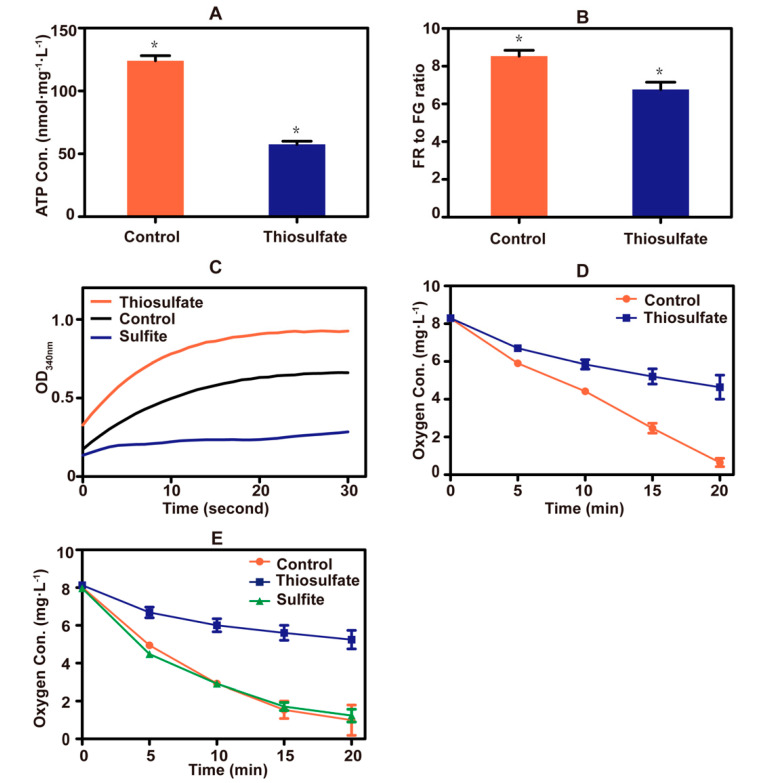
Thiosulfate perturbs the bioenergetics. (**A**) The ATP concentrations of yeast cells. ATP in cell extracts was measured with firefly luciferase. (**B**) The mitochondrial membrane potential of yeast cells. The fluorescent probe JC-1 was used, and a high ratio of red fluorescence (RF) to green fluorescence (GF) represented a high mitochondrial membrane potential. (**C**) The GAPDH activity of yeast cells. The activity was monitored via NADH production. (**D**) Thiosulfate inhibition on oxygen consumption in PBS buffer with glucose addition. The sulfur-starved yeast cells were resuspended in sterile PBS buffer containing 2% glucose. Control, no thiosulfate addition; thiosulfate, + 10 mM thiosulfate. (**E**) Thiosulfate inhibition on oxygen consumption in PBS buffer with TMPD/ascorbate addition. The sulfur-starved yeast cells were resuspended in sterile PBS buffer containing 12.5 mM ascorbate and 1.4 mM TMPD. Control, no thiosulfate addition; thiosulfate, + 10 mM thiosulfate; sulfite, + 10 mM sulfite. The data are averages with standard deviations from three replicates for Figure 5A,B,D,E. * indicates *p* values < 0.01.

**Figure 6 antioxidants-10-00646-f006:**
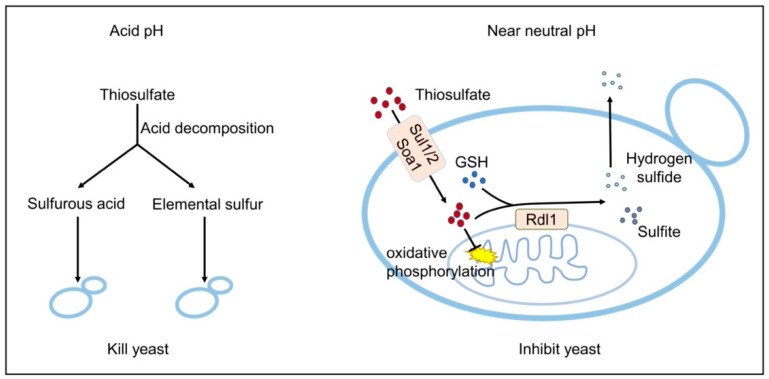
The proposed mechanisms of thiosulfate toxicity to *S. cerevisiae.*

**Table 1 antioxidants-10-00646-t001:** The strains and plasmids used in the study.

Strains and Plasmids	Relevant Characteristics
***S. cerevisiae* BY4742**	*MATα his3*Δ*1 leu2*Δ*0 lys2*Δ*0 ura3*Δ*0*
***E. coli* DH5a**	*supE44, AlacU169, hsdR17, recA1, endA1, gyrA96, thi-1, relA1*
****Δ*sul1 *Δ*sul2 *Δ*soa1***	*SUL1::loxP, SUL2::loxP, SOA1::BLE*
****Δ*rdl1***	*RDL1::loxP*
****Δ*rdl2***	*RDL2::loxP*
****Δ*rdl1 *Δ*rdl2***	*RDL1::loxP, RDL2::BLE*
****Δ*rdl1::RDL1***	*RDL1::loxP,* YEplac195-*RDL1*
****Δ*rdl1::RDL1* C98S**	*RDL1::loxP,* YEplac195-*RDL1* C98S
****Δ*rdl1::RDL2***	*RDL1::loxP,* YEplac195-*RDL2*
**Plasmids**	
***RDL1*-Yeplac195**	*RDL1* in YEplac195, control by own promoter
***RDL1*-Yeplac195 C98S**	*RDL1* C98S in YEplac195, control by own promoter
***RDL2*-Yeplac195**	*RDL2* in YEplac195, control by own promoter

**Table 2 antioxidants-10-00646-t002:** The primers used in the study.

Names	Sequences	Purpose
**R1 ko F**	ATTCTTTCTCGTTTATTTTCAGGGTTTGTGACTAAGAAACGATATTAAAGCTTCGTACGCTGCAGGTC	Knock out *RDL1*
**R1 ko R**	TACTAGCTTACGAAAATACACAGGGTACATACCTAGAGTATACAAGGCCAATACGCAAACCGCCTCT	
**R2 ko F**	GCGATAACTCTCAACAAATGGAAGCGAGACAGAAGAAAAAGACCAACGCTTCGTACGCTGCAGGTC	Knock out *RDL2*
**R2 ko R**	AAGGTTGTCTATATACAGGATATATCGATTATACTTGTTTCTTTTTGGCCCAATACGCAAACCGCCTCT	
**R1 F**	TATGACCATGATTACGCCATTTTATTGGCGCATAGACAAG	Overexpression of Rdl1
**R1 R**	GTCGACCTGCAGGCATGCATGGGGTGTTCGACTAGGTT	
**R2 F**	TATGACCATGATTACGCCAGAACCATCTGAGTACTCGATT	Overexpression of Rdl2
**R2 R**	GTCGACCTGCAGGCATGCAGAAAAAGTCTGAGAAACGTAAAGT	

**Table 3 antioxidants-10-00646-t003:** Total rhodanese activities in wild type and rhodanese-deletion strains.

Strains	Rhodanese Activity
**Wt**	100%
**Δ** **rdl1**	46.6%
**Δ** **rdl2**	70.3%
**Δ** **rdl1** **Δ** **rdl2**	26.3%

## Data Availability

Data are contained within the article or Appendix A.

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
