# Peer review of "The Mechanisms of Thiosulfate Toxicity against *Saccharomyces cerevisiae"

_antioxidants, 2021, doi:10.3390/antiox10050646_

Round 1

Reviewer 1 Report

Article "The mechanisms of thiosulfate toxicity against Saccharomyces cerevisiae" is of good quality but needs some small improvements:

  1. Introduction: l. 24. reference 4 is about yeast S.cerevisiae, so the sentence "Many heterotrophic bacteria..." should be supplemented with these microorganisms. 
  2. Materials and methods: l. 63 these publications are missing from the reference list
  3. Discussion should be extended. 
  4. l. 262 Lactobacillus studies are not presented in this article, so why do the authors write ."our finding suggests that the Lactobacillus"...

Author Response

Thank you for your comments and valuable suggestions.

  1. Introduction: l. 24. reference 4 is about yeast S. cerevisiae, so the sentence "Many heterotrophic bacteria..." should be supplemented with these microorganisms. 
Response: The citation of Refence 4 (Chen et al. 2018) was removed from here.
  1. Materials and methods: l. 63 these publications are missing from the reference list.

Response: References were added.

  1. Discussion should be extended. 

Response: The first paragraph is divided into two.  A detailed description of Figure 6 is added in the new first paragraph.  The solubility limitation of elemental sulfur becomes a new paragraph (3rd in the discussion) and expanded.  We also expanded the last paragraph to indicate the reversible inhibition of thiosulfate near neutral pH, and the results do not contradict its use in treating cyanide poisoning.  

  1. l. 282 Lactobacillus studies are not presented in this article, so why do the authors write ."our finding suggests that the Lactobacillus"...

Response: Changed to “A US patent has used a Lactobacillus strain with thiosulfate to treat urogenital yeast infections (Nivoliez, 2015). Although the mechanism is not reported, we can speculate that the Lactobacillus strain produces lactic acid and lowers pH that facilitates thiosulfate decomposition to release S0 and H2SO3.”

Reviewer 2 Report

Chen and colleagues address an interesting question in their study, namely whether thiosulphate is toxic for the model organism Saccharomyces cerevisiae and, if so, what the toxicity is based on. The introduction is quite short, but contains all relevant information. The methods are described in a comprehensible way. Only in the quantification of thiosulphate with monobromobiman I would think that some more details on detection techniques are needed. This reagent is classically used for the detection of glutathione. Here I would ask the authors for more details. Also the method for measuring the mitochondrial membrane potential needs some more experimental details to increase the comprehensibility. 
In the results, the authors clearly show a concentration-dependent reduction of growth in the presence of thiosulphate, both in the growth assay and in the drop assay. It is very good that both assays were used. The pH dependence of toxicity is also well shown in both assays. The active uptake of thiosulphate into the cell was credibly shown using a triple mutant. The conclusion that rhodanese is responsible for the metabolisation of thiosulphate is very clever and the corresponding mutant assay is extremely impressive. Influence of thiosulphate on ATP production (measured by luciferase, which is a very strong assay but still has its "pitfalls") and membrane potential (making the ATP assay more powerful) are clear The authors themselves show the influence on oxygen uptake of thiosulphate in yeast cells, which in turn may have an influence on the luciferase-dependent detection of ATP. This is likely to be briefly covered in the discussion and I would ask the authors to add to it accordingly. 
The discussion of the article is brief, an orientation even closer to the experimental results would be desirable and is to some extent expandable. 
All in all, I am convinced that after minor changes this paper can be published very well and will be read with interest by various disciplines. 

Author Response

Chen and colleagues address an interesting question in their study, namely whether thiosulphate is toxic for the model organism Saccharomyces cerevisiae and, if so, what the toxicity is based on. The introduction is quite short, but contains all relevant information. The methods are described in a comprehensible way. Only in the quantification of thiosulphate with monobromobimane I would think that some more details on detection techniques are needed. This reagent is classically used for the detection of glutathione. Here I would ask the authors for more details. Also the method for measuring the mitochondrial membrane potential needs some more experimental details to increase the comprehensibility. 

Response: A) Some details about thiosulfate detection by monobromobimane is added.  “Thiosulfate, sulfite, and sulfide react with mBBr to produce derivatives that can be separated by HPLC and detected with a fluorescence detector (Togawa et al., 1992).”

B) Details on the method to detect mitochondrial membrane potentials were added.

“The cells were used to measure mitochondrial membrane potential by using a fluorescent probe 5,5',6,6'-tetrachloro-l,l',3,3'-tetraethylbenzimidazolo-carbocyanine iodide (JC-1) (Beyotime Biotech, Shanghai, China) (Reers et al., 1991), following the manufacture’s instruction. Briefly, 1 mL of the yeast cell suspension was mixed with an equal volume of the purchased JC-1 staining solution (5 μg/mL), incubated at 30°C for 20 min, rinsed twice with ice-cold HEPES buffer (50 mM, pH 7.0), and suspended in the ice-cold HEPES buffer at OD600nm of 0.5. The fluorescence (Red fluorescence: Ex=556 nm and Em=590 nm; Green fluorescence: Ex=490nm and Em=530 nm) was measured by using a microplate reader (BioTek, Synergy H1). The thiosulfate-treated and untreated yeast cells were tested. In theory, JC-1 exists as monomers that emit green fluorescence in cells with low mitochondrial membrane potential, and it forms aggregates that emit red fluorescence in cells with high mitochondrial membrane potential. A decrease in the ratio of red to green fluorescence indicates a reduction in the mitochondrial membrane potential (Jiang et al., 2018).”

In the results, the authors clearly show a concentration-dependent reduction of growth in the presence of thiosulphate, both in the growth assay and in the drop assay. It is very good that both assays were used. The pH dependence of toxicity is also well shown in both assays. The active uptake of thiosulphate into the cell was credibly shown using a triple mutant. The conclusion that rhodanese is responsible for the metabolisation of thiosulphate is very clever and the corresponding mutant assay is extremely impressive. Influence of thiosulphate on ATP production (measured by luciferase, which is a very strong assay but still has its "pitfalls") and membrane potential (making the ATP assay more powerful) are clear. The authors themselves show the influence on oxygen uptake of thiosulphate in yeast cells, which in turn may have an influence on the luciferase-dependent detection of ATP. This is likely to be briefly covered in the discussion and I would ask the authors to add to it accordingly. 

Response: We added the suggested information in the discussion:

“The accumulated thiosulfate inhibits O2 consumption, which in turn lowers membrane potential and ATP levels (Fig. 5).”

The discussion of the article is brief, an orientation even closer to the experimental results would be desirable and is to some extent expandable. 

Response: The first paragraph is divided into two.  A detailed description of Figure 6 is added in the new first paragraph.  The solubility limitation of elemental sulfur becomes a new paragraph (3rd in the discussion) and expanded.  The lethal effects of sulfite and thiosulfate are compared and discussed in a new paragraph. We also expanded the last paragraph to indicate the reversible inhibition of thiosulfate near neutral pH, and the results do not contradict its use in treating cyanide poisoning.  

All in all, I am convinced that after minor changes this paper can be published very well and will be read with interest by various disciplines. 

Response:  Thank you!. Your valuable suggestions and encouragement are highly appreciated.